# *BRCA* Genes and Related Cancers: A Meta-Analysis from Epidemiological Cohort Studies

**DOI:** 10.3390/medicina57090905

**Published:** 2021-08-30

**Authors:** Yen-Chien Lee, Yen-Ling Lee, Chung-Yi Li

**Affiliations:** 1Department of Oncology, Tainan Hospital, Ministry of Health and Welfare, Executive Yuan, Tainan 700, Taiwan; yc_lee@post.harvard.edu; 2Department of Internal Medicine, National Cheng Kung University Hospital, College of Medicine, Tainan 700, Taiwan; 3Department of Public Health, College of Medicine, National Cheng Kung University, Tainan 700, Taiwan; cyli99@mail.ncku.edu.tw; 4Department of Public Health, College of Health, China Medical University, Taichung 406, Taiwan; 5Department of Healthcare Administration, College of Medical and Health Science, Asia University, Taichung 413, Taiwan

**Keywords:** *BRCA1*, *BRCA2*, meta-analysis, cancer

## Abstract

*Background and Objectives:*
*BRCA1* and *BRCA2* are genes located in different chromosomes that are disproportionately associated with hereditary breast and ovarian cancer syndrome. Their association with other cancers remains to be explored. *Materials and Methods:* We systematically reviewed cohort studies to explore the association of *BRCA 1* and *BRCA2* with various cancers except lung cancer. We searched PubMed, Medline (EBSCOhost) and relevant articles published up to 10 May 2021. The odds ratio, standardised morbidity rate and cancer-specific standardised incidence ratio were pooled together as relative risk (RR) estimates. *Results:* Twelve studies were included for analysis. *BRCA* mutation increased pancreatic and uterine cancers by around 3–5- and 1.5-fold, respectively. *BRCA* mutation did not increase brain cancer; colorectal cancer; prostate, bladder and kidney cancer; cervical cancer; or malignant melanoma. *BRCA2* increased gastric cancer with RR = 2.15 (1.98–2.33). *Conclusion:* The meta-analysis results can provide clinicians and relevant families with information regarding increased specific cancer risk in *BRCA1* and *BRCA2* mutation carriers.

## 1. Introduction

Breast cancer genes *BRCA1* and *BRCA2* play a role in DNA repair and co-localise in the nuclear foci with RAD51, a protein required for homologous recombination within a, probably common, DNA repair pathway [1]. The history of BRCA findings is written in the article ”The Twists and Turns in BRCA’s Path” [2]. *BRCA1* is located in chromosome 17q21, whereas *BRCA2* is located in chromosome 13q12.

*BRCA* genes share a common pathway in DNA repair and are well-known for increasing the incidence of breast and ovarian cancers. The mechanisms of the unequal distribution or predominance of breast and ovarian cancers remain obscure. For example, the Breast Cancer Linkage Consortium (BCLC) did not find an increase in the incidence of haematologic malignancies through *BRCA1* or *BRCA2* mutation except that of lymphoma by *BRCA1* mutation [3,4]. One study reported an increase in haematologic cancers [5]. Another study reported a remarkable rise in laryngeal cancer (relative risk (RR): 7.67) [6], whereas others reported none [4]. Our group previously performed a meta-analysis and found no increase in lung cancer incidence due to *BRCA1* or *BRCA2* gene mutation [7]. A systemic review and meta-analysis found conflicting results for colorectal cancer; one study [8] showed that *BRCA1* increases cancer risk (odds ratio (OR): 1.49, confidence interval (CI): 1.20–1.86) and another study [9] showed that *BRCA1* and *BRCA2* do not increase cancer risk.

Due to the available of poly(ADP–ribose) polymerase inhibitors for *BRCA* mutation cancer patients and the advances of precision medicine, the relationships of other cancers, besides ovary and breast needed to be explored. Only cohort studies were selected because of their high evidence level.

## 2. Materials and Methods

### 2.1. Search Strategy and Data Abstraction

We calculated *BRCA1* and *BRCA2* separately because these genes are located in different chromosomes. Lung cancer incidence was previously studied by our group and therefore not included in the present study [7]. The search procedure was similar to our previous study. PubMed and Medline (EBSCOhost) databases were systematically searched for relevant articles published up to 10 May 2021 using the terms “BRCA, cancer” without language restriction. Additional search methods included a manual review of the reference lists of relevant studies. Inclusion criteria were as follows: (1) the study was published and had extractable information, (2) the participants had *BRCA1* or *BRCA2* mutations and (3) the control groups involved patients without mutations or the general population. Abstracts or posters were not selected because their quality is difficult to evaluate. Studies that grouped *BRCA1* and *BRCA2* mutations as a *BRCA* mutation were not included. When the same patient population was used, the work with the highest patient number was selected. Cohort studies with ascertained *BRCA* mutation carriers and cohort studies involving pedigree analysis were analysed together and the ascertained ones were selected for analysis. The Preferred Reporting Items for Systematic Reviews and Meta-Analyses statement was followed for data extraction. Two reviewers (YCL and YLL) independently examined the titles and abstracts of the publications according to the search strategy. The full texts of all potentially relevant publications were retrieved. The data extracted from each study included the publication year, name of the first author, trial type, number of patients, number of observed cases, number of control cases, OR, standardised morbidity rate (SMR), cancer-specific standardised incidence ratio (SIR) and RR. Only cohort studies were selected to decrease the heterogenicity. The Newcastle–Ottawa Scale was used in the quality assessment. The scale comprises eight items with a total of nine points as follows: representativeness of the cohort, selection of control cohort, ascertainment of exposure, demonstration that outcome of interest is not present at the start of study, comparability (two points, study controls for age and another factor), assessment of outcome, follow-up long enough for outcomes to occur and adequacy of follow-up of cohorts.

### 2.2. Statistics

OR, SMR and SIR were treated as equivalent measures of risks and pooled together as RR estimates. RRs were pooled across studies by using inverse-variance weighted DerSimonian–Laird random-effect models to allow for between-study heterogeneity. I^2^ statistics were used to determine between-study heterogeneity. I^2^ is the proportion of the total variation in the estimated effects for each study due to heterogeneity between studies. Analyses were conducted in Stata (version 12.0; Stata Corp, College Station, TX, USA). Two-sided *p* < 0.05 was considered statistically significant.

## 3. Results

Fifty-five full articles were retrieved from a review of 7091 titles and abstracts in total. Additionally, 22 full articles were obtained through the manual review of the reference lists of relevant articles (Figure 1). From these 77 articles, 65 were excluded because of the following reasons: (1) 4 articles had the same cohorts [6,10,11,12]; (2) 10 articles combined *BRCA1* and *BRCA2*; (3) 10 articles had no control groups; (4) 6 articles were unextractable; (5) 15 studies were series or case studies; (6) 20 studies were irrelevant; and (7) 6 studies were excluded for discussions or news, etc. Finally, 12 articles were selected (Figure 1).

Six studies were conducted in multiple countries [3,4,13,14,15,16,17]. The remaining five studies were conducted in the United States [18], the Netherlands [19], the United Kingdom [20], Israel [21] and South Africa [14] (Table 1). Five cohort studies involved ascertained *BRCA* mutation carriers [13,15,16,18,22]. The remaining cohort involved the pedigree to expand the cohort number by adding the pedigree to the calculation of the risk of being a carrier. One study [14] only calculated *p*-values, and overserved results (cases/expected cases) were treated as the SIR. Four studies focused on females only, and one study focused on males (Table 1).

Regarding the quality of studies, most study outcomes [13,14,15,16] were collected personally, by telephone interview or by mailed questionnaire. One study [18] was secured from the Progeny database during genetic counselling session or from the patient’s medical record and one from medical record or death certification [19,20]. The adequacy of follow-up was reported as a mean of 5.5 years [16] or average person years [3,20] in three studies. Only one study [18] reported an 85% completion of follow-up, whereas one study [4] reported 236 patients lost to follow-up. Most studies used cancer data controlled for age-, sex- and country-specific incidence rates [3,4,15,16,19,20]. The scoring methods are given in Table 2. The highest possible total score is 9. The distribution of total scores in the 12 studies is as follows: 8 (two studies), 7 (four studies), 6 (two studies), 5 (three studies) and 4 (one study; Table 2).

The first score, which is the ”representativeness of the exposed cohort”, was scored as positive if the study was considered truly or somewhat representative of the *BRCA* mutation carriers. The second score, which is the “selection of non-exposed cohort”, was scored as positive if the general population or a population with no *BRCA* mutation was selected. The third score, which is the “ascertainment of exposure”, relates to the measurement of *BRCA* mutation at the start of the study. The fourth score, which is the “demonstration that the outcome of interest was not present at start of the study”, was scored as positive when cancer was not presented initially. The fifth score, which is the “comparability of cohorts on the basis of design or analysis”, was scored according to whether the analysis set the initial age and/or an additional factor as the control variable. The sixth score, which is the “assessment of outcome”, was scored positively when the procedure of cancer confirmation was described in the chart retrieved. The seventh score, which is “was the follow-up long enough for outcomes to occur”, was scored positive if the median follow-up was greater than 5 years, which is considered adequate. The eighth and final score, which is the “adequacy of the follow-up of cohorts”, was scored positively when the follow-up was complete or when the subjects lost to follow-up were less than 20%. ★: score one point, —: no point.

Studies [3,4] grouped laryngeal and thoracic cancers for calculation. One study [21] did not separate Hodgkin lymphoma from non-Hodgkin lymphoma and did not include these cancers in the analysis. Cancer types were systematically drawn separately as follows: head and neck cancer (Figure 2), gastrointestinal cancer (Figure 3), biliary and pancreatic cancer (Figure 4), gynaecologic cancer (Figure 5), urologic cancer (Figure 6), thyroid cancer (Figure 7), cancer of the bone and connective tissue (Figure 8), haematologic malignancies (Figure 9), melanoma (Figure 10) and others (Figure 11). Only one study focused on pharyngeal cancer and showed that *BRCA2* increased the incidence of this cancer (Figure 2). Others showed that *BRCA1* or *BRCA2* mutation does not increase head and neck cancer incidence. *BRCA2* mutation increases oesophageal and gastric cancers based on two and six studies, respectively (Figure 3). *BRCA1* or *BRCA2* mutation can increase biliary and pancreas cancer incidence by around threefold (Figure 4). *BRCA1* or *BRCA2* mutation can marginally increase cervical and uterine cancer incidence. Only one study [13] emphasised endometrial cancer incidence (Figure 5). A sensitivity test showed that *BRCA2* mutation increases the incidence of endometrial cancer grouped with other gynaecological cancers (RR 1.74 [1.17–1.58], data not shown). Most of the studies showed that *BRCA1* and *BRCA2* mutations do not increase urological malignancies (Figure 6). Only one study reported an increase in urological malignancy due to *BRCA1* and *BRCA2* mutations [22]. *BRCA1* and *BRCA2* mutations do not increase thyroid cancer incidence (Figure 7). However, *BRCA1* and *BRCA2* mutations can increase bone cancer incidence based on one and two studies, respectively (Figure 8). *BRCA1* and *BRCA2* mutations can increase leukaemia incidence based on three studies (Figure 9). Only one study related to *BRCA2* focused on uveal melanoma with increased incidence [20]. Others showed that the mutation has no effect on melanoma incidence. Figure 11 shows the incidence of other cancers, including cancers of the peritoneum, intestinal tract, nasal sinuses, pleura, other genital organs, eye and several sites ill-defined in the studies. All these cancers were defined ”cancer excluding breast, ovarian and non-melanoma skin cancer” by studies.

## 4. Discussion

*BRCA*-associated tumours occur disproportionately in women’s breast and ovary, as well as in other body parts, but to a lesser degree. Some studies grouped these other cancers with breast and ovarian cancers and pointed out that *BRCA* mutations increase cancer incidence, which we think might be attributed largely to breast and ovarian cancers.

Figure 2, Figure 3, Figure 4, Figure 5, Figure 6, Figure 7, Figure 8, Figure 9, Figure 10 and Figure 11 are summarised below. Cancer incidence based on one to three studies was considered less strong. These incidences are as follows. *BRCA1* and *BRCA2* mutations increased leukaemia (RR 2.3 [1.21–4.39] and 1.79 [1.03–3.12], respectively). *BRCA1* mutation increased endometrial cancer (RR 5 [3.06–8.16]) and Hodgkin’s disease (RR 3.79 [1.97–7.28]). *BRCA2* mutation increased pharyngeal cancer (RR 7.30 [1.66–32.01]), oesophageal cancer (RR 4.1 [1.26–13.3]), hepatic cancer (RR 3.59 [1.04–12.34]), gallbladder and biliary cancers (RR 4.7 [1.42–15.53]), endometrial cancer (RR 6.2 [2.33–16.52]), uveal melanoma (99.4 [15.69–629.85]) or bone cancer (RR 6.76 [1.09–42.07]) but did not increase cancers of the tongue or salivary glands or the small intestines. *BRCA1* mutation did not increase the incidence of oesophageal cancer, liver cancer, gallbladder and bile duct cancer or bone cancer. *BRCA* mutations were not related to cancers of the buccal cavity, pharynx and oral cavity; cancers of the larynx and throat; colon or rectum cancer; cancers of the urogenital system or urinary bladder testis; haematological malignancy; other lymphoma; myeloma; thyroid cancer; or connective tissue disease.

Meta-analyses based on more than three studies were as follows. *BRCA* mutation increased pancreatic cancer by three- to fivefold and other uterine cancers by around 1.5-fold. *BRCA2* mutation increased gastric cancer incidence (RR 2.15 [1.98–2.33]), whereas *BRCA1* did not increase gastric cancer incidence. *BRCA* mutation did not increase the incidences of brain cancer, colorectal cancer, prostate cancer, bladder and kidney cancer, cervical cancer or malignant melanoma.

One study [23] stated that *BRCA* mutations are associated with the increased incidence of endometrial carcinoma. *BRCA1* and *BRCA2* mutations were put together for analysis; most studies were case-control studies, and cases were selected amongst patients with endometrial carcinoma. Two studies [23,24] were from the same author (Lavie O). Another study [24] related to meta-analysis grouped both *BRCA* genes together. Our study, as well as another meta-analysis study [9], did not show that *BRCA* mutations increase colorectal cancer. Mok Oh et al. [8] showed that *BRCA1* mutation is associated with colorectal cancer with an OR of 1.49 (1.200–1.86) but not *BRCA2* mutation. However, the studies by Thompson and Easton and Brose and Ford used data from the BCLC for calculation. Gene studies use many resources, and they were conducted in medical centres or cooperation groups. We tried to avoid repeat sampling by selecting the same author name or institute only once, and the larger population was selected. Studies on genetic influences also use many resources. Therefore, the population usually overlaps. For example, three studies selected patients from the University of Toronto [13,15,16]. We could not tell them apart and grouped them together. Prostate cancer incidence did not increase in our study, nor in that of Mok Oh et al. [8]. However, they [25] later performed another meta-analysis and showed that prostate cancer incidence is increased by *BRCA1* or *BRCA2* mutation. The studies buy Thompson and Ford from the BCLC were selected for *BRCA1* analysis. Case-control studies were selected as well. More possible repetitive inclusions may have occurred if more studies were included. Additionally, most of the studies gathered data in person or by telephone interview. The ascertained diagnosis of cancer by recall or during follow-up was hard to confirm. Therefore, the designs of the study were all described as cohort study (Table 1).

Studies [18] regarding the direct testing of *BRCA* mutations and variants showed that these mutations marked increase ovarian cancer with SIR values of 139.12 (*BRCA1*) and 74.93 (*BRCA2*), whereas the effects in other cancers mostly have SIR values of around 1–2. Most studies excluded patients with both mutations [18], but one study [16] included all patients without specifying the calculation method.

The effects of both genes in different cancers should be separately calculated for specificity when explaining to patients during daily practice; moreover, many variants remain largely unknown. Patients with strong family history may be screened for possible mutations; however, some patients may give less attention to family history and not undergo screening. Furthermore, patients with non-breast or non-ovarian cancer might be less likely to come for DNA testing. One study [19] tried to minimise this variance by using a population with a less-selected mutation. However, we cannot evaluate these affects.

*BRCA* gene mutations can lead to cancer. The classification systems used in the studies are highly heterogeneous, and variants of uncertain clinical relevance add a challenge to interpretations. Some variants were initially regarded as pathogenic but ended up having no clinical significance, whereas others were the opposite [26]. Many studies discussed the clinical importance of *BRCA1* and *BRCA2* variants. These findings need a consensus, and a long way is still required to reach a conclusion. Nevertheless, some studies [22] implied that specific mutations, for example, the *BRCA2* exon 11 mutation, increase the incidence of colorectal, stomach and pancreatic cancers more than other mutations. Some studies also included pedigree to enlarge the study population, which might have led to selection bias and misclassification. These could be improved in the initial designs of direct sequence genes and with solid information on direct gene effects. The *BRCA1* gene spreads over 81 kb, whereas *BRCA2* is 84 kb long. Determination of whether point mutation or deletion leads to clinically significant tumour incidence changes is time-consuming. Sensitivity analysis studies separated the results in carriers from those in pedigrees (data not shown). However, this approach is the best we could do with the evidence given by the current cohort studies to provide an initial guide.

## 5. Conclusions

*BRCA1* and *BRCA2* mutations increased the incidence of pancreatic cancer (RR 3–5) and uterine cancer (RR 1.5) but were not associated with brain cancer, colorectal cancer, prostate cancer, bladder and kidney cancer, cervical cancer or malignant melanoma. Patients with these mutations might pay attention to pancreatic and uterine cancers, with emphasis on gastric cancer, besides breast and ovarian cancer.

## Figures and Tables

**Figure 1 medicina-57-00905-f001:**
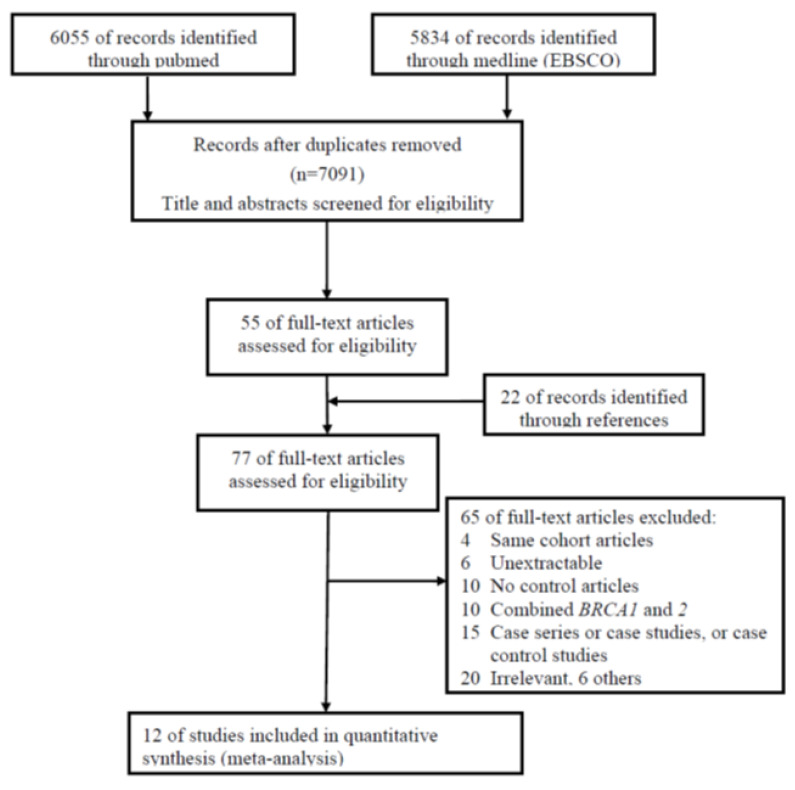
Study selection scheme.

**Figure 2 medicina-57-00905-f002:**
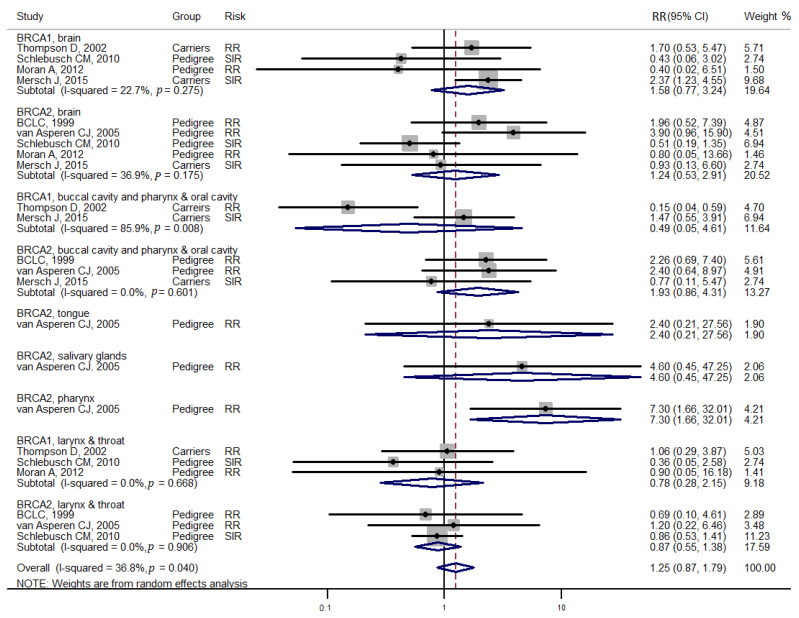
Meta-analysis of the association between *BRCA* gene mutation and head and neck cancer incidence. CI: confidence interval, RR: relative risk, SIR: cancer-specific standardised incidence ratio. Solid diamonds denote ratio point estimate from each study, open diamonds represent pooled overall results and the dashed line denotes the overall pooled point.

**Figure 3 medicina-57-00905-f003:**
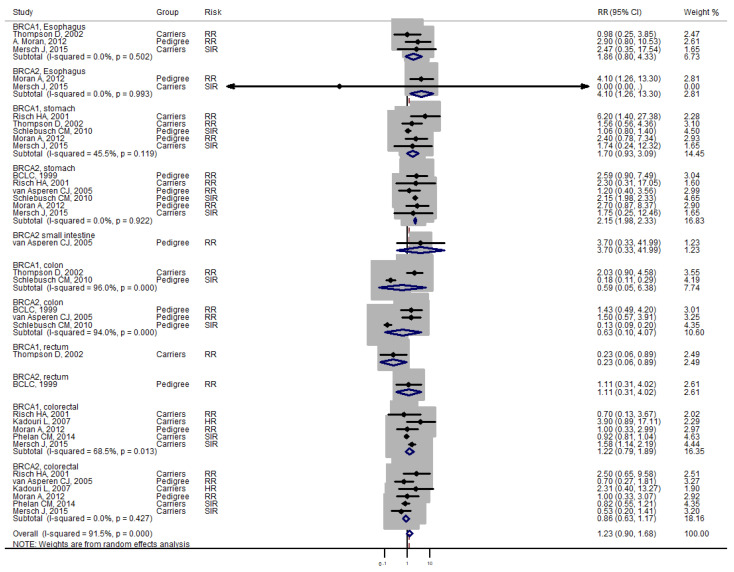
Meta-analysis of the association between *BRCA* gene mutation and gastrointestinal cancer incidence. CI: confidence interval, RR: relative risk, SIR: cancer-specific standardised incidence ratio. Solid diamonds denote ratio point estimate from each study, open diamonds represent pooled overall results and the dashed line denotes the overall pooled point.

**Figure 4 medicina-57-00905-f004:**
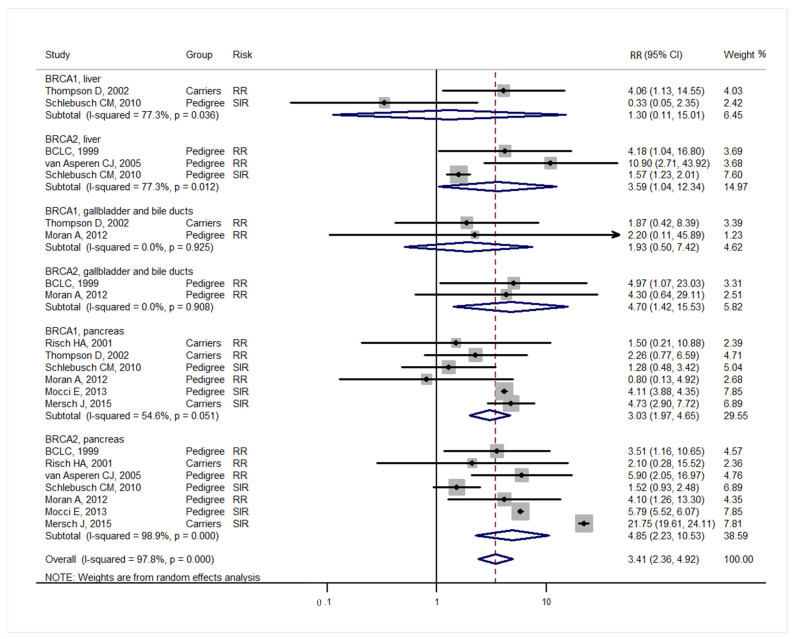
Meta-analysis of the association between *BRCA* gene mutation and biliary and pancreatic cancer incidence. CI: confidence interval, RR: relative risk, SIR: cancer-specific standardised incidence ratio. Solid diamonds denote ratio point estimate from each study, open diamonds represent pooled overall results and the dashed line denotes the overall pooled point.

**Figure 5 medicina-57-00905-f005:**
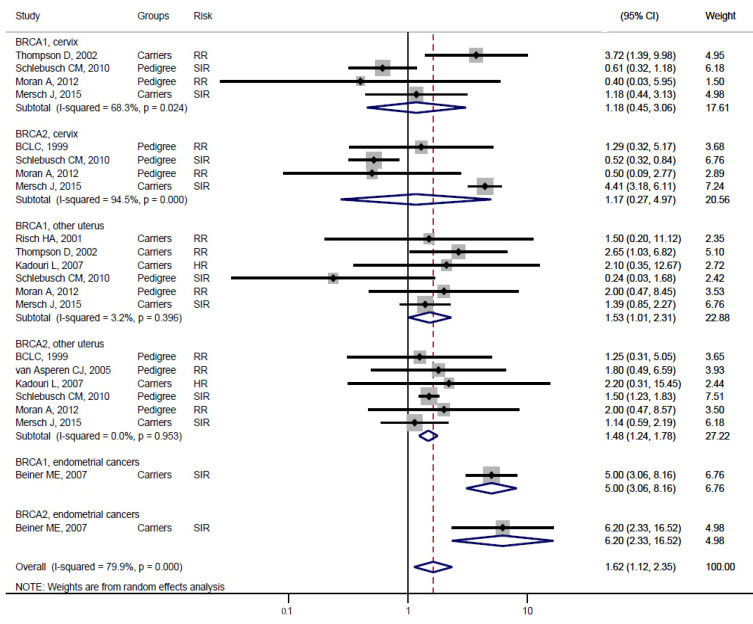
Meta-analysis of the association between *BRCA* gene mutation and the incidence of uterine and cervical cancers. CI: confidence interval, RR: relative risk, SIR: cancer-specific standardised incidence ratio. Solid diamonds denote ratio point.

**Figure 6 medicina-57-00905-f006:**
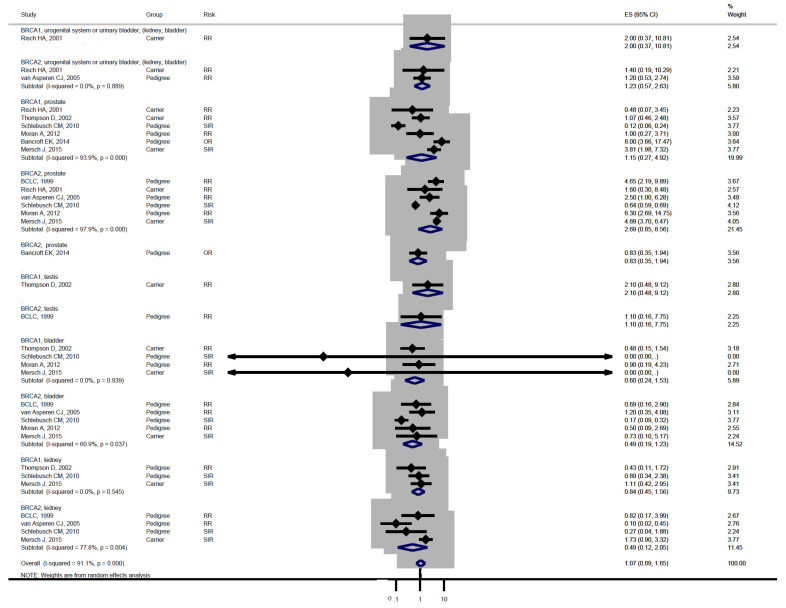
Meta-analysis of the association between *BRCA* gene mutation and urologic malignancy incidence. CI: confidence interval, RR: relative risk, SIR: cancer-specific standardised incidence ratio, OR: odds ratio. Solid diamonds denote ratio point.

**Figure 7 medicina-57-00905-f007:**
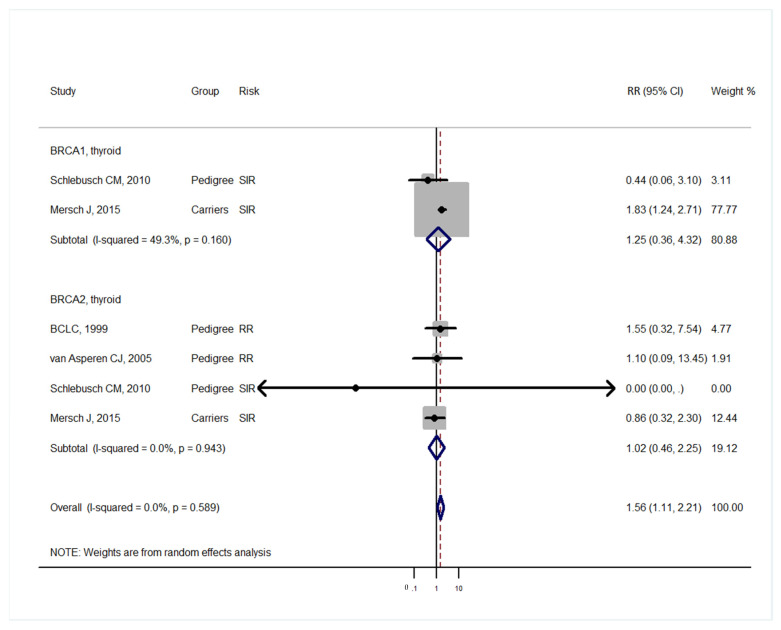
Meta-analysis of the association between *BRCA* gene mutation and thyroid cancer incidence. CI: confidence interval, RR: relative risk, SIR: cancer-specific standardised incidence ratio. Solid diamonds denote ratio point.

**Figure 8 medicina-57-00905-f008:**
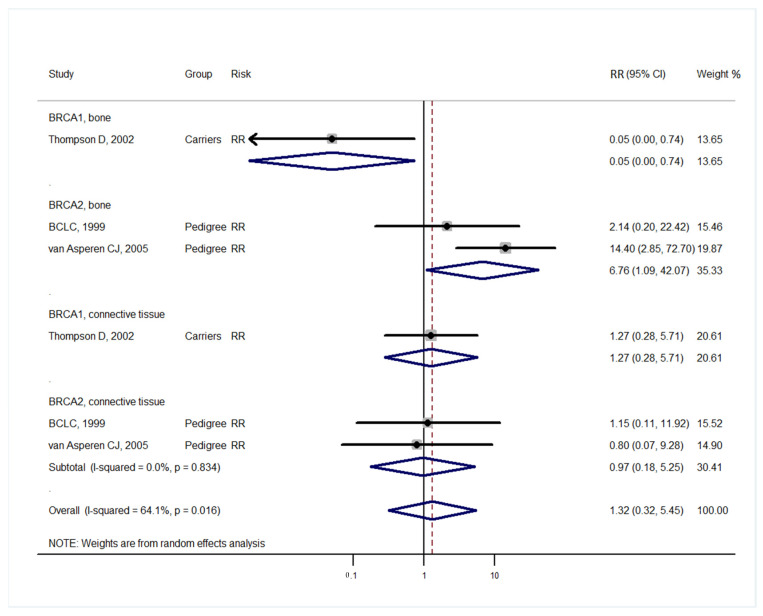
Meta-analysis of the association between *BRCA* gene mutation and the incidence of bone and connective tissue cancers. CI: confidence interval, RR: relative risk, SIR: cancer-specific standardised incidence ratio. Solid diamonds denote ratio point.

**Figure 9 medicina-57-00905-f009:**
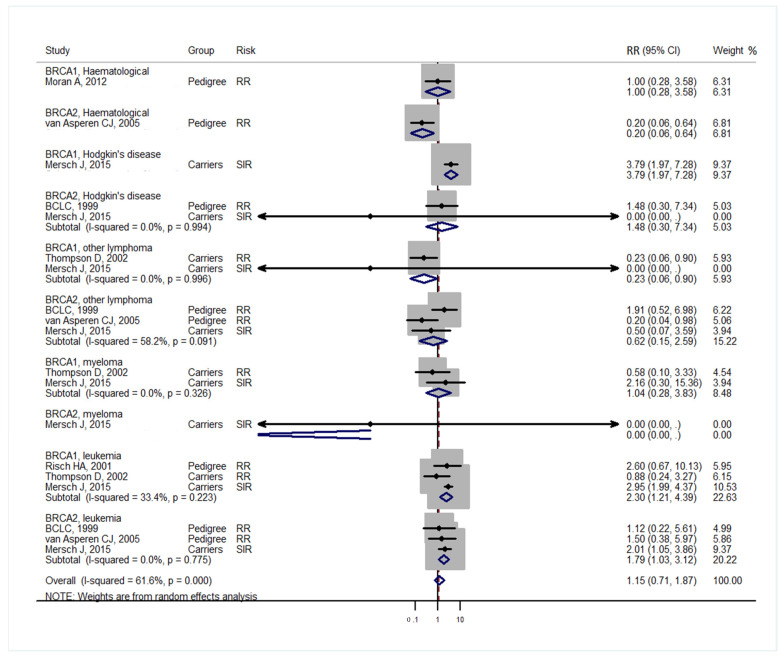
Meta-analysis of the association between *BRCA* gene mutation and haematologic malignancy incidence. CI: confidence interval, RR: relative risk, SIR: cancer-specific standardised incidence ratio. Solid diamonds denote ratio point.

**Figure 10 medicina-57-00905-f010:**
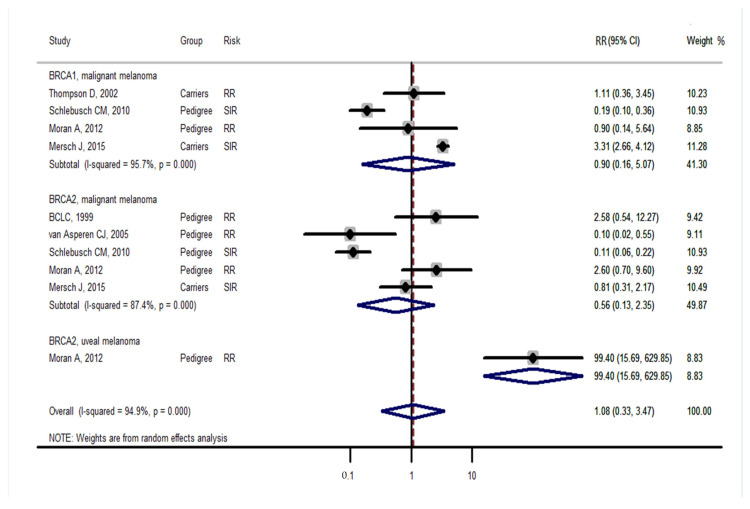
Meta-analysis of the association between *BRCA* gene mutation and melanoma incidence. CI: confidence interval, RR: relative risk, SIR: cancer-specific standardised incidence ratio. Solid diamonds denote ratio point.

**Figure 11 medicina-57-00905-f011:**
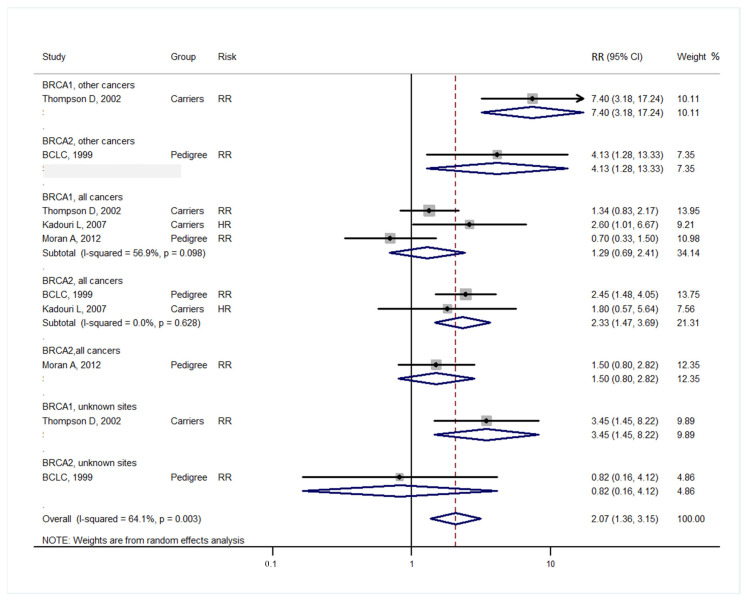
Meta-analysis of the association between *BRCA* gene mutation and the incidence of other cancers. CI: confidence interval, RR: relative risk, SIR: cancer-specific standardised incidence ratio. Solid diamonds denote ratio point. Other cancers include cancers of the peritoneum, intestinal tract, nasal sinuses, pleura, other genital organs, eye and several sites ill-defined in the studies; all cancers, excluding breast, ovarian and non-melanoma skin cancers.

**Table 1 medicina-57-00905-t001:** Summary of the baseline characteristics of the included studies.

Studies	Study Design	BRCA Carrier	Mean Age (Range)	Sex Distribution	Control	Median Follow-Up	Country (Population)
BCLC, 1999 [4]	Retrospective cohort	Kindred *BRCA2* 3728 Pedigree studies	-	-	Cancer incidence in five continents	-	BCLC, Western Europe, the US and Canada
Risch HA, 2001 [22]	Cohort studies	Direct sequencing	*BRCA1* 51.2 *BRCA2* 57.5	Female only	Relatives of cases not carrying mutation	-	Toronto
Thompson D, 2002 [3]	Cohort study	Direct sequencing *BRCA* 2245	-	-	Cancer incidence in five continents	26.12 years (P-Y)	BCLC, Western Europe, the US and Canada
van Asperen CJ, 2005 [19]	Retrospective cohort	Pathogenic *BRCA2* mutation 50% prior probability (pedigree)	-	1088 (56%)/803 (44%)	Dutch cancer incidence rates	-	Netherlands
Beiner ME, 2007 [13]	Prospective cohort	Direct sequencing *BRCA1* 619 *BRCA2* 236 Both 2	54.4 (45–70)	Female only	Cancer incidence in five continents	3.3 y	University of Toronto, multicentre North American, Europe and Israel
Kadouri L, 2007 [21]	Cohort studies Ashkenazi	*BRCA1* 229 *BRCA2* 100 Excluded both mutations	-	Female only	Ashkenazi non-carrier	52.8, 57 years (P-Y)	Hadassah Medical Centre (Jerusalem, Israel)
Schlebusch CM, 2010 [14]	Cohort study	Pedigrees	-	3682 (all), 57% female	National Cancer Registry	-	Clinic at the University of Pretoria, South African families
Moran A, 2012 [20]	Prospective cohort	Pedigree included *BRCA1* 1,15 *BRCA2* 1526	-	1100/715 931/595	Northwest of England (1975–2005)	22.6 years (P-Y)	Regional Genetics Clinics at Manchester and Birmingham (Northwest and West Midlands in England)
Mocci E, 2013 [15]	Cohort study	Deleterious *BRCA1* and *BRCA2* mutations and pedigree *BRCA1* 11,946 *BRCA2* 7773	-	6490/5457 4271/3502	Cancer incidence in five continents	-	Six centres in the United States (Northern California Breast Cancer Family Registry, New York site of the BCFR, Utah site of the BCFR, Philadelphia site of the BCFR), Canada (Ontario Familial Breast Cancer Family Registry) and Australia (Australian Breast Cancer Family Registry).
Phelan CM, 2013 [16]	Prospective	BRCA deleterious mutation *BRCA1* 5481 *BRCA2* 1474 Both 60	47.3 (30–74)	Female only	Cancer incidence in five continents	5.5 years (mean)	Five countries from Canada, the United States or Europe
Bancroft EK, 2014 [17]	Prospective cohort	50% inheriting a mutation, pedigree *BRCA1* 791 *BRCA2* 731	60.1 59.8	Male only	Family member test negative	≥5 years	Multicentre, the Royal Marsden Hospital NHS Foundation Trust, St Mary’s Hospital, Manchester; The Princess Anne Hospital, Southampton; and Addenbrooke’s Hospital, Cambridge.
Mersch J, 2015 [18]	Prospective	BRCA deleterious mutation, variants included Excluded both mutations	Mean age last contact 49.36 (range, 17–90 years)	584 (57.82)/29 (46.77) 426 (42.18)/33 (53.23)	United States Cancer Statistics: 1999–2010 Incidence and Mortality Web-Based Report	-	MD Anderson Cancer Centre

**Table 2 medicina-57-00905-t002:** Methodologic quality of studies based on the Newcastle–Ottawa Scale (*N* = 12).

Studies	Representativeness of the BRCA Gene	Selection of Non-BRCA or General Population	Ascertainment of BRCA	Demonstration That Cancer Was Not Present Initially	Study Controls for Initial Age and/or an Additional Factor	Assessment of Outcome	Was Median Follow-Up 5 Years or More?	Adequacy of Follow-Up (>80%)	Total
BCLC, 1999 [4]	★	★	★	★	★★	—	—	★	7
Risch HA, 2001 [22]	★	★	★	★	——	—	—	—	4
Thompson D, 2002 [3]	★	★	★	★	★★	★	★	—	8
van Asperen CJ, 2005 [19]	★	★	★	★	★★	★	—	—	7
Beiner ME, 2007 [13]	★	★	★	★	★—	—	—	—	5
Kadouri L, 2007 [21]	★	★	★	★	——	—	★	—	5
Schlebusch CM, 2010 [14]	★	★	★	★	★—	—	—	—	5
Moran A, 2012 [20]	★	★	★	★	★ ★	★	★	—	8
Mocci E, 2013 [15]	★	★	★	★	★★	—	—	—	6
Phelan CM, 2014 [16]	★	★	★	★	★★	—	★	—	7
Bancroft EK, 2014 [17]	★	★	★	★	——	★	★	—	6
Mersch J, 2015 [18]	★	★	★	★	★—	★	—	★	7

★: score one point, —: no point.

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
