# Peer review of "BRCA Genes and Related Cancers: A Meta-Analysis from Epidemiological Cohort Studies"

_medicina, 2021, doi:10.3390/medicina57090905_

Round 1
Reviewer 1 Report
Many of the conclusions about the pathophysiological activity of a gene or protein depend on the analyzes conducted by high-dimensional data. Therefore, an important component of the meta-analysis is the coherence of biological effects resulting from the omics data.
The omics data, which characterize biomedicine in the last 20 years, for their complex and intrinsically heterogeneous nature, for the wide range of different experimental designs, for the possible and many usable technologies and for the many methods of bioinformatics analysis, represent a set of high-dimensional experimental data, massively archived and very insidious, where a meta-analysis can lead to distorted results.
BRCA genes 1 and 2 produce proteins involved in DNA error repair and their mutations alter these processes. The pathological activities driven by these mutations are many and not yet well known. The very biology of tumors is unknown. What we struggle to know is "how", "where" and "when" mutant genes and their coded products (proteins) exert their functional activities in the cell and tissues through specific molecular driving mechanisms. There is a lack of knowledge of all the space-time references that characterize the metabolic actors and their behavior in a tumor cell. Only today we have some shy approach (Next Generation Sequencing) to get some answers.
For many years, we have collected data from thousands of cancer patients in the study centers on the world. This information, appropriately anonymized, is shared in public databases that researchers from all over the planet can access. It is an immense mass of data in which precious information is hidden. Bioinformatics, through its computational analysis methodologies, can help to find, organize, classify and interpret them. These vast masses of data today represent the preferred study material in cancer research and are organized in informatics systems called biomedical BigData. The experimental design with which these data are produced, analyzed and studied depends only on the investigating laboratory.
Today, biomedical BigData represents the laboratory in which many research groups operate producing experimental data and extracting information to be used in routine scientific work to study the physiopathological mechanisms that underlie almost all modern human diseases.
Many of these data have an original sin, they lack space-time references. The "how", the "when" and the "where" are missing. This makes the data and analyzes conducted on them very questionable and insidious and requires careful and specific examination when doing meta-analyzes.
The logical-experimental approaches used in today's biomedical research on breast cancer and BRCA 1 and 2 mutations are no exception. Many works on the breast cancers have experimental designs that are based on the logic illustrated. The facts say that breast cancers can be surgically treated in early stages, but when this doesn’t happen, the number of deaths in a short time is very high. The fundamental reason is that we do not know the molecular mechanisms used by mutated proteins to define how, where and when they act. This explains why there is still no effective cure. Therapeutic interventions without a clear understanding of pathogenesis are often ineffective or even harmful.
The results got with these approaches are published in thousands of papers that meta-analyzes try to dissect and classify to extract hidden information or make systematic reviews.
A similar and striking example is cellular hepatocarcinoma (HCC), one of the most deadly cancers. It is without cure, with a great analogy with breast cancers for what concerns development, progression, survival, and scientific knowledge. There are thousands of scientific papers in PubMed on HCC which, when analyzed, seem sound but when compared they reveal different experimental designs and analyzes with profound methodological errors, many false positives. The result is that to date hundreds of different biomolecules are known that each single research group deems the perfect therapeutic target: too many for a single tumor. So, you don't get very far. Also with HCC the space-time references are missing.
A discussion point that I don’t wish to develop is all this information accumulate in Big Data systems. Thus, BigData can give biased or flawed results if the archived data is not error free (according to the Veracity law about Big Data). We consider that predictions of personalized medicine require the use of artificial intelligence on data arising from procedures described above. Thus the probability of errors is concrete and personalized predictions can lead to Medical Errors in daily clinical practice.
BRCA 1 and 2 are no exception. They come from high-dimensional data from which we generate scientific and clinical predictions that aim at treatments, drugs or personalized medicine but can have intrinsic sources of errors.
At present, biomedical Big Data, AI, and personalized medicine are a combination based on marketing, and the speed of markets, but very little on science and correct scientific knowledge.
The authors are well aware of these real difficulties and declare: "But this is the best we could do from current cohort studies based evidence to provide the suffering a guide". This referee disagrees because it must be clearly explained to readers where are the causes that make meta-analysis difficult and what are the possible approaches to overcome these problems.
Therefore, I ask the authors to discuss meta-analysis approaches through which they can control and filter the heterogeneity of the studies and inherent errors that can bias studies and data coming from BRCA 1 and 2 researches. I do not seem to have found any trace of the verification of the validity and truthfulness of the primary data of these researches.
Here are a few illustrative links of what I have said:
https://academic.oup.com/bib/article/18/4/602/2562766
https://www.ncbi.nlm.nih.gov/pmc/articles/PMC7847983/
https://pubmed.ncbi.nlm.nih.gov/29763131/
https://www.ncbi.nlm.nih.gov/pmc/articles/PMC6313275/
https://jcp.bmj.com/content/71/11/995.long
Author Response
Many of the conclusions about the pathophysiological activity of a gene or protein depend on the analyzes conducted by high-dimensional data. Therefore, an important component of the meta-analysis is the coherence of biological effects resulting from the omics data.
The omics data, which characterize biomedicine in the last 20 years, for their complex and intrinsically heterogeneous nature, for the wide range of different experimental designs, for the possible and many usable technologies and for the many methods of bioinformatics analysis, represent a set of high-dimensional experimental data, massively archived and very insidious, where a meta-analysis can lead to distorted results.
BRCA genes 1 and 2 produce proteins involved in DNA error repair and their mutations alter these processes. The pathological activities driven by these mutations are many and not yet well known. The very biology of tumors is unknown. What we struggle to know is "how", "where" and "when" mutant genes and their coded products (proteins) exert their functional activities in the cell and tissues through specific molecular driving mechanisms. There is a lack of knowledge of all the space-time references that characterize the metabolic actors and their behavior in a tumor cell. Only today we have some shy approach (Next Generation Sequencing) to get some answers.
- Thank you for the reviewer comment. The initiation of this topic is due to one of my patients who had double cancer of breast and lung. She also had BRCA mutation as well. The cause of her death was lung cancer progression. That’s why we think this topic is important to investigate whether BRCA genes drive any specific cancer, at least from epidemiology view. So patients won’t spend lots of many by selling house or borrowing from the bank to pay for BRCA gene directed treatment target therapy. The underlying mechanism we believe will be revealed in the near future.
For many years, we have collected data from thousands of cancer patients in the study centers on the world. This information, appropriately anonymized, is shared in public databases that researchers from all over the planet can access. It is an immense mass of data in which precious information is hidden. Bioinformatics, through its computational analysis methodologies, can help to find, organize, classify and interpret them. These vast masses of data today represent the preferred study material in cancer research and are organized in informatics systems called biomedical BigData. The experimental design with which these data are produced, analyzed and studied depends only on the investigating laboratory.
Today, biomedical BigData represents the laboratory in which many research groups operate producing experimental data and extracting information to be used in routine scientific work to study the physiopathological mechanisms that underlie almost all modern human diseases.
Many of these data have an original sin, they lack space-time references. The "how", the "when" and the "where" are missing. This makes the data and analyzes conducted on them very questionable and insidious and requires careful and specific examination when doing meta-analyzes.
- We agree with the reviewer points. This is an imperfect world. But by being unable to do nothing, we can at least do something from this imperfect data with the hope of some benefit to the patients.
The logical-experimental approaches used in today's biomedical research on breast cancer and BRCA 1 and 2 mutations are no exception. Many works on the breast cancers have experimental designs that are based on the logic illustrated. The facts say that breast cancers can be surgically treated in early stages, but when this doesn’t happen, the number of deaths in a short time is very high. The fundamental reason is that we do not know the molecular mechanisms used by mutated proteins to define how, where and when they act. This explains why there is still no effective cure. Therapeutic interventions without a clear understanding of pathogenesis are often ineffective or even harmful.
- By saying to do nothing, we tried to do the piece by piece little by little with the hope of approaching the whole pictures steps by steps.
The results got with these approaches are published in thousands of papers that meta-analyzes try to dissect and classify to extract hidden information or make systematic reviews.
A similar and striking example is cellular hepatocarcinoma (HCC), one of the most deadly cancers. It is without cure, with a great analogy with breast cancers for what concerns development, progression, survival, and scientific knowledge. There are thousands of scientific papers in PubMed on HCC which, when analyzed, seem sound but when compared they reveal different experimental designs and analyzes with profound methodological errors, many false positives. The result is that to date hundreds of different biomolecules are known that each single research group deems the perfect therapeutic target: too many for a single tumor. So, you don't get very far. Also with HCC the space-time references are missing.
- Thank you for the reviewers teaching about HCC and makes us think more.
A discussion point that I don’t wish to develop is all this information accumulate in Big Data systems. Thus, BigData can give biased or flawed results if the archived data is not error free (according to the Veracity law about Big Data). We consider that predictions of personalized medicine require the use of artificial intelligence on data arising from procedures described above. Thus the probability of errors is concrete and personalized predictions can lead to Medical Errors in daily clinical practice.
- Thank you for the reviewer comment. We are on the way and try to make the aim correct to benefit the patients.
BRCA 1 and 2 are no exception. They come from high-dimensional data from which we generate scientific and clinical predictions that aim at treatments, drugs or personalized medicine but can have intrinsic sources of errors.
At present, biomedical Big Data, AI, and personalized medicine are a combination based on marketing, and the speed of markets, but very little on science and correct scientific knowledge.
The authors are well aware of these real difficulties and declare: "But this is the best we could do from current cohort studies based evidence to provide the suffering a guide". This referee disagrees because it must be clearly explained to readers where are the causes that make meta-analysis difficult and what are the possible approaches to overcome these problems.
- We have pointed out some difficulties in the studies. For example page 13 of 21 line 388 “Mutations of BRCA genes lead to cancer are well known. The classification systems are highly heterogeneous and variants of uncertain clinical significance are challenges in interpretations. Some were initially regard pathogenic come out of no clinical significant while others went to the opposite direction[26]. “ The other difficulties are page 13 of 21, line 394 “To make thing even more complicated, some study [22] implicated that specific mutation, for example BRCA2 exon 11 mutation had excess cancer incidence in colorectal, stomach, pancreatic, etc., than other mutations.” Also, regarding the selection population groups were not the same had been pointed out. “some studies included pedigree to enlarge the studies population which might lead to selection bias and misclassification.” It’s hard to overcome all these problem unless the initial study designs have change as well as all the mutations are clear.
Therefore, I ask the authors to discuss meta-analysis approaches through which they can control and filter the heterogeneity of the studies and inherent errors that can bias studies and data coming from BRCA 1 and 2 researches. I do not seem to have found any trace of the verification of the validity and truthfulness of the primary data of these researches.
- We have added this sentence “These could be improved in the initial designs of direct sequence genes and with solid information of direct gene effects. The BRCA1 gene spreads over 81 kb, whereas BRCA2 is 84 kb long. Determination if point mutation or deletion leads to clinical significant tumor incidence changes are time consuming. Sensitivity analysis studies separated the results in carrier from those in pedigree (data not shown). However, this approach is the best we could do in the evidence given by the current cohort studies to provide an initial guide. “ in page 13 of 23.
Reviewer 2 Report
This is a meta-analysis review of BRCA genes in association to various cancers. The authors aim to study involvement of BRCA genes in various cancers and only use published literature to do so. Although the manuscript presents some value to the readers, it is largely inadequate in the present form.
Major concerns of the manuscript are listed below-
- Referencing must be improved throughout the manuscript. One example, in line 38 ref (1) does not comment on BRCA2 at all. Therefore, a separate reference for BRCA2 must be included. All other references in the manuscript must be checked for such errors.
- Lines 42-45 are confusing. What are the dissimilarities between BRCA genes and APC and TP53? The authors should mention this clearly.
- Apart from PARP inhibitors mentioned in line 59 what are the other reasons to select BRCA genes for this study? The authors should elaborate upon this in the introduction.
- Out of the 77 articles found, 65 were excluded and the authors have cited the reasons for doing so. The authors should explain in greater detail the significance of 'not' including the other articles. For example, the say- "20 studies were irrelevant". This is incomplete and should be explained in detail. Same goes for other exclusion criteria as well.
- Figures 3 and 6 seem compressed and not clear. The style is not consistent with other figures.
- The English throughout the manuscript needs to be improved. Discussion particularly is very confusing. Few examples are-i. Lines 343-345 needs grammatical correction. ii- line 351 "they" is unnecessary. iii= line 353-354 needs rephrasing. iv- lines 358-361 are very confusing and need to be rephrased to clearly express the authors thoughts. The message is unclear. v- line 365 needs correction. These are just few errors in the language but more careful phrasing is necessary for the manuscript to make sense. I would highly recommend help from professionals. I would recommend taking professional help if possible.
Author Response
Reviewer 2:
This is a meta-analysis review of BRCA genes in association to various cancers. The authors aim to study involvement of BRCA genes in various cancers and only use published literature to do so. Although the manuscript presents some value to the readers, it is largely inadequate in the present form.
Major concerns of the manuscript are listed below-
- Referencing must be improved throughout the manuscript. One example, in line 38 ref (1) does not comment on BRCA2 at all. Therefore, a separate reference for BRCA2 must be included. All other references in the manuscript must be checked for such errors.
We thank for the reviewer comment and have checked that all the citations errors were in Table 1 and 2 and had correct all the references. Thank you very much.
- Lines 42-45 are confusing. What are the dissimilarities between BRCA genes and APC and TP53? The authors should mention this clearly.
We have deleted APC and TP53 to make it clear. Thank you very much.
- Apart from PARP inhibitors mentioned in line 59 what are the other reasons to select BRCA genes for this study? The authors should elaborate upon this in the introduction.
We have state that “advances of precision medicine” in line 60 as well. We focus on BRCA gene is due to one of my patients suffered from both lung cancer and breast cancer with BRCA gene mutation. After our previous study had shown that BRCA gene might not be a drive mutation of lung cancer (from epidemiology data) which cause her death. So she didn’t have to spend lots of money to buy the target therapy drugs. There are currently PARP inhibitor for BRCA genes treatment. In the near future, we think therapy will be flourish. But this is story behind and we think its too redundant to put it here.
- Out of the 77 articles found, 65 were excluded and the authors have cited the reasons for doing so. The authors should explain in greater detail the significance of 'not' including the other articles. For example, the say- "20 studies were irrelevant". This is incomplete and should be explained in detail. Same goes for other exclusion criteria as well.
Some articles initially screening by the eyesight might think there are treasures in it but only to open it to find out that this is irrelevant. For example, PMID 23233716 (Prevalence and type of BRCA mutations in Hispanics undergoing genetic cancer risk assessment in the southwestern United States: a report from the Clinical Cancer Genetics Community Research Network J Clin Oncol. 2013 Jan 10;31(2):210-6.), from the title, we think it might be related but only to download it to find out not related. We have attached our Excel file of the search PMID number to classify for your references. If we put more detailed in it, the article will be too long and too confusion and cannot be directly to the points.
- Figures 3 and 6 seem compressed and not clear. The style is not consistent with other figures.
It’s consistent but too large. We have upload the original file separately and hope it will be clearer.
- The English throughout the manuscript needs to be improved. Discussion particularly is very confusing. Few examples are-i. Lines 343-345 needs grammatical correction. ii- line 351 "they" is unnecessary. iii= line 353-354 needs rephrasing. iv- lines 358-361 are very confusing and need to be rephrased to clearly express the authors thoughts. The message is unclear. v- line 365 needs correction. These are just few errors in the language but more careful phrasing is necessary for the manuscript to make sense. I would highly recommend help from professionals. I would recommend taking professional help if possible.
We had rephrase the words “ meta-analysis based on studies…” in line 341. We have change line 341 to 346 as following “Meta-analysis based on studies more than three were as following. BRCA mutation increased pancreatic cancer around 3 to 5 fold, other uterus around 1.5 fold. BRCA2 increased stomach cancer [RR 2.15 (1.98 to 2.33)]. BRCA1 didn’t increased stomach cancer incidences. BRCA mutation didn’t increase brain cancer, colorectal cancer, prostate, bladder and kidney, cervical cancer and malignant melanoma.” And hope it’s clearer.
We have deleted “they”. In line 351 as reviewer suggestd.
Line 358 to 361 had been rephrase that “Genes studies took lots of resources and were generally conducted in medical centers or cooperation groups. We tried to avoid repeat sampling by selecting same authors name or institute only once. And the larger population was selected. “
Line 365, we have rephrase “Prostate cancer incidences were not increased in our studies same as previous Mok Oh et al study[8].”
We have sent for English Proofreading. Please see certification. Thank you very much.
I cannot upload my search excel file. Could only put some part of it here.
|
